# Biological Activity of Complexes Involving Nitro-Containing Ligands and Crystallographic-Theoretical Description of 3,5-DNB Complexes

**DOI:** 10.3390/ijms25126536

**Published:** 2024-06-13

**Authors:** Daniela Fonseca-López, Johan D. Lozano, Mario A. Macías, Álvaro Muñoz-Castro, Desmond MacLeod-Carey, Edgar Nagles, John Hurtado

**Affiliations:** 1Grupo de Investigación en Química Inorgánica, Catálisis y Bioinorgánica, Departamento de Química, Universidad de los Andes, Bogotá 111711, Colombia; d.fonseca100@uniandes.edu.co; 2Crystallography and Chemistry of Materials, Departamento de Química, Universidad de los Andes, Bogotá 111711, Colombia; jd.lozanoc@uniandes.edu.co (J.D.L.); ma.maciasl@uniandes.edu.co (M.A.M.); 3Facultad de Ingeniería, Arquitectura y Diseño, Universidad San Sebastián, Bellavista 7, Santiago 8420524, Chile; alvaro.munozc@uss.cl; 4Inorganic Chemistry and Molecular Materials Laboratory, Instituto de Ciencias Aplicadas, Facultad de Ingeniería, Universidad Autónoma de Chile, El Llano Subercaseaux 2801, Santiago 8910124, Chile; desmond.macleod@uautonoma.cl; 5Facultad de Química e Ing. Química, Universidad Nacional Mayor de San Marcos, Lima 15081, Peru

**Keywords:** biological activity, coordination complexes, nitrobenzoic acid, crystallographic data, computational calculations

## Abstract

Drug resistance in infectious diseases developed by bacteria and fungi is an important issue since it is necessary to further develop novel compounds with biological activity that counteract this problem. In addition, new pharmaceutical compounds with lower secondary effects to treat cancer are needed. Coordination compounds appear to be accessible and promising alternatives aiming to overcome these problems. In this review, we summarize the recent literature on coordination compounds based on nitrobenzoic acid (NBA) as a ligand, its derivatives, and other nitro-containing ligands, which are widely employed owing to their versatility. Additionally, an analysis of crystallographic data is presented, unraveling the coordination preferences and the most effective crystallization methods to grow crystals of good quality. This underscores the significance of elucidating crystalline structures and utilizing computational calculations to deepen the comprehension of the electronic properties of coordination complexes.

## 1. Introduction

Nitro-containing ligands and their corresponding complexes are valuable compounds for different applications due to their biological properties and interesting structural architectures. These remarkable properties have been studied from a computational perspective. Particularly, a large number of reports make reference to the antimicrobial and anticancer activities of these complexes, in part as a response to some issues observed in other organic compounds, such as side effects and drug resistance [1,2,3,4,5,6,7,8,9,10]. In this context, this work focuses on nitro-containing ligands, for instance, 3,5-dinitrobenzoate (3,5-DNB), which has been demonstrated to be a widely used versatile ligand, as it shows a diversity of potential coordination modes through carboxylate and nitro groups toward metal ions, allowing for the formation of coordination complexes with different and enhanced properties [2,3]. Considering the structure–property relationship, the structural variety observed in these complexes is frequently reported using spectroscopic techniques. However, X-ray single-crystal data are considered a reliable source of information to characterize the different coordination modes on these complexes [2]. Following this tendency, we revised the crystallographic information files found in the database. As a result, multiple crystal structures are reported in the Cambridge Crystallographic Data Centre (CCDC) database with compounds containing the 3,5-DNB ligand. It is interesting that despite the possibility of nitro and carboxylic groups acting as coordinating groups, the information reported in the CCDC suggests that there is a higher probability for the carboxylic group to dominate the coordination to 3d metals and some rare earth elements [4,9]. Considering this behavior, the design of polymers or polynuclear molecules containing this ligand should anticipate that nitro groups tend to participate in the supramolecular structure of crystals by interactions involving physical forces instead of forming chemical bonds, which is expected even in solution. However, some exceptions are known in the literature for metals such as Na [10,11], K [6], Cs [12], Ag [6], and Tl [13]. The growing of single crystals to elucidate the molecular structures of these complexes is a tedious procedure. However, abundant information regarding the usual crystallization strategies is found in the literature. In this work, the experimental factors that influence the nucleation are discussed, such as the basicity hardness of crystallization solvents, temperature, and reaction time. In addition, the structural behavior of these complexes is studied using computational methods such as density functional theory (DFT), which allows for the exploration of the electronic and spectroscopic properties, accounting for additional information such as quantitative structure–activity relationships (QSARs). In this work, we present a thorough review showing complexes formed by nitro-containing ligands accompanied by discussions regarding the biological activity, structural diversity properties supported by computational calculations.

## 2. Nitrobenzoic Acid (NBA)

Nitrobenzoic acid and its derivatives are essential materials in pharmaceutical industries [14,15,16]. A literature study revealed a good deal of work on the properties of NBA and its substituted derivatives [17]. Similarly, benzoate ligands are used in bioinorganic chemistry because the carboxylate group exhibits different coordination modes [18], which allows them to be widely used in the synthesis of complexes with potential biological activity due to the synergy between the NBA-derived ligand and the metal [2]. Additionally, the nitro groups have a reducing capacity since they can generate free radicals that cause cell damage (oxidative stress). Also, they can interact with biomolecules that are fundamental for the survival of different microorganisms. The diverse properties of the nitro group are due to its high reactivity. Among these, the nitro group is a strong electron withdrawer, reducing the electron density within its compounds through inductive and resonance effects. Thus, it can undergo reactions with nucleophiles. The nitro group, by reduction, can be converted to nitroso, oxime, and amino groups [19,20,21,22].

## 3. Biological Activity of NBA Derivative Complexes

In the context of the biological activity of complexes with NBA, many natural products containing the nitro group present significant biological activities, such as antibiotics, antifungals, insecticides, and antitumors [18,19,20,21,22,23]. Therefore, a wide variety of drugs containing nitro group(s) have been developed, and thousands of nitrogenous compounds were prepared and screened against various diseases in the 1940s. Many nitro compounds are employed to treat different diseases, for instance, Parkinson’s disease [24]. The mechanism of action has not been completely elucidated since it may vary depending on the structural differences of the compounds.

Most nitro compounds have the mechanism of enzymatic bioreduction based on the formation of free radicals intoxicating bacterial and parasite cells [25], being the nitroreductase (NTR) enzymes responsible for catalyzing this reduction. Specifically, type I NTRs are not oxygen-sensitive and execute a two-electron reduction, while type II NTRs are oxygen-sensitive and achieve a one-electron reduction (Figure 1) [26,27]. As an example, the trypanocidal activity of drugs is decided by the NTRs within the parasite, which is not found in humans.

### 3.1. Antifungal and Antibacterial Activity of NBA Derivative Complexes

Currently, there is a great variety of strains that present resistance to the antibiotics or fungicides used today in therapeutic treatments. This fact has become a problem for human health sanitary organizations such as the World Health Organization (WHO) [28]. Microorganisms are associated with dangerous infections in humans. Bacteria such as *Staphylococcus aureus*, *Escherichia coli*, and *Pseudomonas aeruginosa* are the most common, and *Candida albicans* and *Aspergillus niger* are the fungal strains that cause hospital-acquired diseases [29].

This resistance is due, in part, to the restricted number of drugs for the treatment of these infections and the lack of selectivity of these commercial drugs [30]. For these reasons, it has been of great interest to study coordination complexes, considering that they have been examined against bacteria and fungi with enhanced results against these pathogens. The higher antimicrobial activity of these complexes compared to free ligands is promoted by five features, which are as follows: (a) the chelate effect, (b) the charge of the complex, (c) the character of the coordinated ligands, and (d) the nuclearity of the metal center in the complex [31,32]. In general terms, the most important is the chelate effect, since it causes the complexes to be more lipophilic than the ligand. After chelation, the reduced polarity of the complex is caused by the ligand orbital overlapping and the delocalization of the positive charge of the metal center. Therefore, the lipophilicity of the chelate and the complex increases [1], contributing to the penetration into the lipid membrane, allowing for the attack of the enzymes and DNA of microorganisms (Figure 2) [33].

In this context, antibacterial drugs based on coordination complexes are necessary to control the development of bacterial and fungal resistance. Among these studied compounds, cobalt(II) complexes containing dinitrobenzoate (DNB) ligands, have shown interesting preliminary results against *Candida albicans* species, leading to further preclinical assays for compounds **1**–**7** in Figure 1 [18,34,35].

In 2007, copper(II) complexes containing the ligands 4-chloro-3-nitrobenzoic acid (4-Cl-3-NB) and 1,3-diamino propane (1,3-DAP) or o-phenylenediamine (o-PDA) were reported. The complex [Cu(4-Cl-3-NB)(o-PDA)]Cl exhibits activity against *Enterococcus faecalis*, *Pseudomonas aeruginosa*, *Staphylococcus aureus*, and *Escherichia coli*. Compound **8** in Figure 2 presents antibacterial activity against *Staphylococcus aureus* and *E. faecalis* with a MIC of 39 µg/L. This result is related to the redox potential of the complex due to its cationic nature, probably mediated by the bidentate mode of complexation of the carboxylate group [36,37]. In the same way, three hybrid complexes (inorganic-organic), [Cu(DIEN)(4-NB)_2_]·H_2_O (**9**), [Cu(DIEN)_2_](4-NB)_2_ (**10**), and [Cu(DIEN)(4-NB)(H_2_O)](4-NB) (**11**) (where 4-NB = 4-nitrobenzoate, DIEN = diethylenetriamine) (Figure 2) were synthesized at room temperature, exhibiting antibacterial activity against *Pseudomonas aeruginosa*, *Escherichia coli*, *aureus Klebsiella pneumoniae*, and *Staphylococcus*, due to a wider inhibition halo around the wells compared to the control solvent DMSO [38].

Furthermore, the in vitro antibacterial activity of the ligand, 4-NB, and the complexes [Co(4-NB)_2_(H_2_O)_4_]·2H_2_O (**12**) and [Ni(4-NB)_2_(H_2_O)_4_]·2H_2_O (**13**) (Figure 2) were screened against *Staphylococcus aureus*, *Klebsiella pneumonia*, and *Bacillus subtilis* (gram-positive) and *Streptococcus mutans* and *Escherichia coli* (gram-negative) bacteria and compared with gentamycin and clotrimazole as antibiotic and antifungal standards, respectively. In this case, the complexes presented higher activity than the free ligand but lesser activity than gentamycin against all bacteria, except Staphylococcus aureus, at a concentration of 100 μg/mL. Additionally, the ligand and the complexes were more active than clotrimazole for *Candida albicans* and *Aspergillus niger* [39].

Silver(I) complexes generally present linear to octahedral geometries, which generate complexes with versatile structures. In addition, silver complexes with Ag-S, Ag-N, or Ag-O bonds or with ligands derived from organic acids or amines have been shown to exhibit biological activity. Accordingly, two silver complexes with 3-nitrobenzoate ligands were reported. The complexes [Ag_2_(3-NB)_2_(DAB)] (**14**) and [Ag_2_(3-NB)_2_(DAC)] (**15**) (Figure 3) ((3-nitrobenzoate (3-NB), 1,2-diaminocyclohexane (DAC), and 1,4-diaminobutane (DAB)) were screened against *Candida albicans*, *Escherichia coli*, and *Staphylococcus aureus*. The complexes have better activity than free ligands and show superior activity to tetracycline against *Staphylococcus aureus* and *Escherichia coli*. Also, the IC_50_ results showed that complexes present effective activity against *Candida albicans*, whereas tetracycline and free ligands exhibit negative activities [33].

As a remarkable point, Ibragimov et al. reported in 2017 the synthesis of three different types of metal complexes with ligands diethanolamine (DEA) and 2-nitrobenzoic acid (2-NB) [40]. All complexes were screened against *Fusarium oxysporum* and *Candida albicans* fungi. The free ligands DEA and 2-NB inhibited the growth of fungi at the screened concentration (1.5 mg/mL). However, in the case of the complexes, complex **18** (Figure 4) with the mixed ligands configuration shows the highest activity, which is due to the synergy of the two ligands and the nickel metal center. Nonetheless, in complexes **16** and **17** (Figure 4), the biological activity was also high; therefore, all these screened compounds showed excellent activity against *Candida albicans*. Compounds **19** and **20** (Figure 4), synthesized using 4-NB and methanolamine (MEA), were screened against *Fusarium oxysporum* and *Aspergillus niger*. Monoligands MEA and 4-NB present fungicidal action. However, the activity observed in the complexes is higher, being even more elevated for the mixed ligand binuclear complex **20**, having an antimicrobial activity against *Aspergillus niger* with a zone of inhibition of 30 mm at a concentration of 0.25 mg/mL [41,42]. From these results, it is possible to conclude that, to achieve a better enhancement in the activity, mixed-ligand metal complexes could represent an interesting option because these may provide higher biological activity due to the synergistic activity of both ligands complemented by the toxicity of metal ions in the pathogen.

### 3.2. Anticancer Activity of NBA Derivative Complexes

Cancer treatment has long been a worldwide problem in human health. Anticancer agents, metal-derivatives, are currently the most useful chemotherapeutic drugs and have been applied clinically for years [43]. For example, cisplatin, oxaliplatin, and carboplatin are used in the chemotherapy of several types of cancer and bind to the DNA of cancer cells to disrupt their replication and consequently stop cancer cell division (Figure 3).

DNA adducts turn on apoptotic routes that cause tumor damage and cell death. Additionally, these agents alter the respiratory chain by altering the function of mitochondria, leading to the production of ROS (reactive oxygen species). Also, oxaliplatin and cisplatin bind to mitochondrial DNA and interfere with its replication and transcription [44]. These mechanisms are the most common for metal complexes and affect the G0/G1 phase. However, when chemotherapy using cisplatin is performed, adverse effects such as ototoxicity, nephrotoxicity, or electrolyte disturbances occur [45,46]. Therefore, the idea of designing new metal-based anticancer drugs to improve treatment efficacy and reduce toxicity is an attractive research idea.

In the pursuit of new antitumor drugs, Mallick et al. highlighted rhenium as a potential metal center due to its lower toxicity compared to other heavy metals. They isolated a few dirhenium complexes containing different NBA ligands, particularly the dirhenium (II, III) complexes. Thus, it is important to study the reactivity of the dirhenium centers bridged by different nitrobenzoate ligands to evaluate their stability in complexes **21**-**23** (Figure 5). Their activity was evaluated against cancer cells finding that they act in a similar way to cisplatin, binding to DNA and inhibiting its replication and protein synthesis [47].

The cytotoxicity of complexes **21**–**23** to breast cancer cells was determined using the MTT assay. The number of viable cancer cells was observed to decrease with increasing complex concentration. The IC_50_ values for **21**–**23** confirm that nitrobenzoate complexes are more active than free ligands toward tumor cell lines. Furthermore, a morphological study by flow cytometry concluded that morphological features detected were caused by cell death, hence a quantitative decrease in the G0/G1 phase [47]. In the same way, Pb(II) (Figure 6) complex **24** with 4-NB as a ligand was synthesized and presented a tetranuclear Pb_4_O_4_-cubane structure. The use of lead complexes is unusual; however, Pb(II) complexes are attractive in their biological activities, such as binding to DNA [48], anticancer [49], and antimicrobial properties [50]. The MTT assay to determine the anticancer activity against cell lines was used, and Complex **24** showed the best activity against cancer cells but showed lower inhibitory action than the *cis*-platinum drug [51].

Moreover, other metals have been used to synthesize new drugs against cancer, such as copper(II), manganese(II), and nickel(II). Zeng et al. recently synthesized copper(II) complexes using 4-Cl-3-NB as the main ligand and 1,10-phenantroline (**25**, **26**) (Figure 7). The complexes decreased the proliferation of human cancer cells, and Complex **25** had greater antitumor activity than cisplatin. Additionally, an evaluation of the cell cycle by flow cytometry showed that these complexes likewise arrested the cell cycle in the G0/G1 phase, similar to the dirhenium complexes [52]. Additionally, a study showed that Mn(II), Ni(II), and Cu(II) complexes with 2-Cl-5-NBA and heterocyclic compounds induced the apoptosis of human lung and colon cancer cells. In the case of lung cancer cells, it was observed that complexes **28** and **29** (Figure 7) similarly exhibited 47.09 and 40.26% growth-suppressive activity at 60 μM, respectively. However, complex **27** (Figure 7) showed 75.70% suppression (IC_50_ = 8.82 μM) at 20 μM. When colon cancer cells were used, it was observed that complexes **27** and **28** (Figure 7) showed similar growth activity, resulting in 72.70% (IC_50_ = 0.00053 μM).

## 4. Coordination Behavior of the 3,5-DNB Ligand and Crystallization Methods of the Complexes: A Perspective from the Crystal Structures in the CCDC Database

To help to better understand the properties of the complexes and their possible applications, it is very important to know the crystalline structure, and this knowledge is extended by means of computational calculations on molecular structure. For this reason, in the following two sections, the results in these areas focused on coordination behavior of the 3,5-DNB ligand, crystallization methods of the complexes, and calculations of the structures of NBA derivative complexes are expanded.

Considering the wide use of the 3,5-DNB ligand in the synthetic routes to obtain complexes containing 3d metals with interesting properties, in this section, we present a discussion based on the crystal structures reported in the CCDC database. From a structural perspective, 3,5-DNB is known as a useful ligand that exhibits a range of coordination modes involving carboxylate and nitro groups. This diversity of coordination allows for the use of this ligand in the assembly of molecular systems and coordination polymers and even helps in the building of interesting supramolecular architectures through intermolecular interactions [6]. The potential of carboxylate and nitro groups to form coordination bonds is illustrated in the molecular electrostatic potential (MEP) analysis of 3,5-dinitrobenzoic acid (3,5-DNBA) with similar negative potential values for both groups, which can be interpreted as potential Lewis bases that are comparable in magnitude [53]. The formation of interesting structural frameworks by the participation of both groups is observed in the crystal structures of compounds containing 3,5-DNB with Na [10,11], K [6], Cs [12], Ag [6], and Tl [13] (Figure 8). However, the connectivity with 3d transition metals is rather different, and coordination bonding involving nitro groups is quite uncommon.

An exploration in the CCDC database version 5.41 (date of search, June 2023), for coordination compounds containing the 3,5-DNB ligand with 3d transition metals, leads to interesting results. Crystal structures containing Cu metal are the most common with approximately 50 reported structures, followed by Co (40 structures), Zn (30 structures), Ni (29 structures), Mn (13 structures), Fe (1 structure), V (1 structure), and zero reports containing Cr, Ti, and Sc. Considering Cu-complexes, the great majority of coordination bonds occur using the carboxylate group (Figure 8) in a monodentate or bidentate manner and form bridging bonds that are controlled by inductive, resonance, polarizability, and steric effects [54], including the possibility for intermolecular hydrogen bonds that stabilize the crystal packing (Figure 8 and Figure 9).

As an exception to the CCDC database overview, the crystal structure of the compound Catena-[(μ-3,5-dinitrobenzoato)-(μ-hydroxo)-copper] shows the contribution of one nitro group in the coordination of Cu atoms with an O-Cu bond distance of 2.59 Å, which is long compared to the observed lengths when carboxylate is the coordination group (1.9–2.1 Å) (CCDC: LIYVOG01) (Figure 8 and Figure 10) [7]. This coordinating behavior of nitro groups is also observed in the compound Catena-[(μ2-ⴄ^2^-azido)-(μ2-3,5-DNB)-copper], where the participation of this fragment helps in the formation of a 3D framework structure with an O-Cu bond distance of 2.61 Å (CCDC: YIXXAG) (Figure 8) [55]. Considering the few examples (only two from 50) containing a nitro group coordinating the Cu metal, it is interesting that the CCDC-LIYVOG compound was obtained from a reaction in a Teflon-lined stainless-steel autoclave heated for 45 h at 423 K, while the CCDC-YIXXAG compound was obtained from a simple mixture of reactants at room temperature and ambient pressure, ruling out a possible effect of the synthesis method employed on the coordinating behavior of nitro groups. Considering the other 48 reported structures containing Cu atoms, it is possible that this behavior could be influenced by a co-ligand-solvent effect. A similar behavior was observed in the compound Catena-((μ2-3,5-DNB)-(3,5-DNB)-(1H-imidazole)-aqua-nickel(II)) (CCDC: UXEQAQ) with an O-Ni bond distance of 2.60 Å. Considering all the crystal structures reported in the Cambridge Crystallographic Data Centre, CCDC, these are the only three examples where nitro groups participate in the coordination bonds, which allows us to conclude that, in the case of 3d transition metals, the 3,5-DNB ligand usually forms coordination bonds through the carboxylate group, and the nitro groups prefer to form intermolecular interactions in the packing, such as hydrogen bonds, nitro···nitro, and van der Waals interactions [2,5].

The coordination bonding of nitro groups in the crystal structures of CCDC-LIYVOG01, CCDC-YIXXAG, and CCDC-UXEQAQ is crucial to obtain polymeric structures when no other ligands in the structure, apart from 3,5-DNB, have this potentiality (Figure 8). However, due to the low frequency of such nitro interactions in the literature for 3d transition metals, it is not convenient to rely on these interactions to synthesize polymeric structures. Instead, other polymers can be achieved using co-ligands, or even the bridging characteristics of 3,5-DNB can help in its formation (Figure 10). This is the case for 3,5-DNB-forming compounds with the lanthanoid series. The CCDC database reports the crystal structures containing the complete group, except for Pm and Lu. In all structures, 3,5-DNB tends to form coordination bonds through the carboxylate group, which acts in monodentate, bidentate, and bridging forms. Some examples are observed in the structures, and their CCDC codes are summarized in Table 1.

In all lanthanoid structures, 3,5-DNB tends to form coordination bonds through the carboxylate group, which act in monodentate, bidentate, and bridging forms. The preference for this sort of coordination in 3d and lanthanide elements can be explained by the electronic availability of oxygen atoms in carbonate and nitro groups. The inductive effect, electronegativity, and resonance of electropositive nitrogen affect the electronic structure of oxygen atoms in the nitro groups, which does not occur in carboxylates; hence, in these groups, the oxygen’s electrons are freer and localized to form a coordination bond. As mentioned before, the bridging forms are important in the formation of polymeric structures (Figure 11). This is the case for the CCDC-PEZQUJ structure in which the carboxylate group periodically bridges two Eu centers. Nonetheless, polymer formation with 3,5-DNB is not always a consequence of bridging formation, as occurs in the case of the CCDC-BOSDIY complex, which is a dinuclear molecular structure of Ho. Moreover, the π-stacking interaction of aromatic rings in 3,5-DNB is also important in the growth of polymer complexes, as in the case of CCDC-EXAVAB containing Gd.

Additionally, the interactions of the 3,5-DNB ligand with 3d transition metals can lead to mono-, di-, or trinuclear molecular compounds. Computational studies allow us to observe that the presence of adequate co-ligands, the electronic nature of the metallic center, and the effect of the Lewis basicity hardness of solvents of crystallization represent the driving force to synthesize a desired compound, where, as demonstrated, the mono-, di-, or trinuclear nature can influence the properties in the obtained compounds (Figure 12) [2,5,59]. 

Considering the demonstrated importance of the synthesis conditions in the formation of crystal structures of varied characteristics, several synthesis methods have been used to prepare a great variety of coordination and polymeric complexes, including 3,5-DNB as a precursor. Hydrothermal and self-assembly solution methods are the most used. Table 2 summarizes some of these complexes synthesized by hydrothermal synthesis.

The general method of this synthesis consists of preparing a mixture of the metal salt (chlorides or acetates) and the different ligands in water (hydrothermal) or organic solvent (solvothermal) and then adding the solution in a Teflon reactor. As shown in Table 2, three important factors must be considered: temperature, time, and Teflon reactor volume. The synthesis temperature, time, and reactor volume of all complexes were approximately 120–150 °C, 3–4 days, and 25 mL, respectively. These parameters seem to be very important for the crystallization process. The synthesis of coordination complexes is challenging, especially if suitable single crystals are desired. In this sense, the pH of the solutions has a demonstrated impact on the results. The synthesis of [Ni_2_(3,5-DNB)_4_(bpy)_2_(H_2_O)] and [Co_2_(3,5-DNB)_2_(bpy)_2_] is conducted while keeping the pH value at approximately 6 through the addition of NaOH [60]. In addition, in the production of three different Cu(II) isomers with a general formula of ([Cu(dtcd)(3,5-DNB)_2_]_n_) through one-pot hydrothermal synthesis, a pH between 3.5 and 4.5 allows for the formation of a mixture of all three isomers. However, only presents were obtained when the pH was kept between 4.5 and 6.0, and at pH 2.5–3.5, only one isomer was detected [61]. The synthesis of two 1D copper (II) coordination polymers [Cu_4_(μ3-OH)_2_(dtb)(3,5-DNB)_6_(H_2_O)_2_]·2H_2_O (1) and Cu(dtb)(3,5-DNB)_2_ (2) [dtb = N, N ′-bis(4H-1,2,4-triazole)butanamide] also seems to be dependent on the pH of the solutions. At low pH values (2–3), pure crystals of polymer 2 are obtained. However, if the pH values are increased to 3.0–4.5, the yield of 1 increases, while the yield of 2 decreases [71]. With these results, the literature suggests that hydrothermal synthesis is highly pH-dependent. Nevertheless, the main problem with this method is related to the yield since it is very difficult to overcome 70%, as shown in Table 2. 

Considering this last remark, other alternatives can be implemented to improve the yielding problem. The self-assembly solution method is one of the most common syntheses used to prepare coordination complexes due to its great yields and synthetic control. The general process consists of dissolving the metal salts (chlorides, acetates, or sulfates) and the different ligands in an organic solvent, e.g., methanol, with or without temperature control under constant stirring. Through slow solvent evaporation, single crystals are obtained. Table 3 summarizes some of these coordination complexes synthesized by the self-assembly solution method. 

Table 3 shows better yields for the self-assembly solution method compared to the solvothermal method. Despite the improved results, it is important to carefully control the synthesis conditions, especially when single crystals are desired. In the case of the compound [Co(3,5-DNB)_2_(py)_2_(H_2_O)_2_], variations from room temperature to 55 °C influence the crystallization of the polymorphs with space groups C2/*c* and P2_1_/*c*, respectively [82]. As discussed before, the crystallization solvents are important for the building of the crystal structures of these coordination complexes. When 3,5-DNBA and 4,4′-bypiridine (bpy) ligands react with a source of Co(II), depending on the mixture of solvents, different structures are obtained using the same conditions.

At room temperature, [Co(3,5-DNB)_2_(bpy)_2_(CH_3_O)_2_] (1) was obtained in methanol (CH_3_OH) and [Co(3,5-DNB)_2_(bpy)_2_(CH_3_O)_2_]CH_3_COCH_3_ (2) in a methanol/acetone mixture, with the CH_3_OH molecule coordinating the metallic center in 1 and behaving as the crystallization solvent in 2 [83]. Additional examples are observed when [Cu(3,5-DNB)_2_(Py)_2_] is recrystallized in DMSO/H_2_O, forming [Cu(3,5-DNB)_2_(DMSO)_2_(H_2_O)_2_] [84], and when [Co(3,5-DNB)_2_] is recrystallized in H_2_O, CH_3_OH, EtOH: acetone, and DMSO, producing mononuclear complexes, while acetone and acetonitrile trigger the synthesis of trinuclear compounds (Figure 12).

## 5. Theoretical Study of 3,5- and 2,4-Dinitrobenzoate Metal Complexes

Density functional (DF) modeling has been extensively used to obtain insights into the structural and electronic properties of monomers and coordination polymers, accounting for their quantitative structure–activity relationship (QSAR) [34], magnetic [59,85,86], and luminescent properties [87,88]. Such evaluation is useful to rationalize their biological activity [28,89] and catalytic performance [90,91], among other applications. 3,5-DNBA acts as a useful organic building unit connecting different metal centers in an extended array, leading to 3D supramolecular networks [92]. Such a unit has been studied theoretically in terms of vibrational, electronic, and thermodynamic properties [93], denoting intramolecular C–N–O⋯H–C and C–O⋯H–C weak interactions, which can also be included in an overall extended array. The formation energies of the resulting extended area can be estimated via DFT calculations, which allows for the rationalization of the favored polymorph to be observed in the molecular solid resulting from the coordination polymer, as evidenced by experimental thermal analysis, denoting the preference between concomitant phases, which prefer an ionic charge arrangement [94]. The stability between coexisting phases is evaluated experimentally via thermal analysis accounting for their densities, suggesting higher stability for the species with larger melting points. On the other hand, 3D high-energy-density and low-sensitivity materials can be obtained, such as [Cu(3,5-DNBA)(N_3_)] [55].

Jassal et al. synthesized seventeen complexes of 3,5- and 2,4-dinitrobenzoate (2,4-DNB) involving alkali, alkaline, and different transition metals, which were structurally characterized by X-ray diffraction, among other techniques [82]. This leads to 0-dimensional monomers, paddle-wheel dimers, pseudocubane, helices, and ladders. In addition, linear one-dimensional tapes, pseudiodiamonds, and brick-wall-type two- and three-dimensional networks show the rich structural versatility, novelty, and thermal stability provided by 2,4-DNB in the formation of inorganic–organic hybrid networks. It must be remarked that the 3,5-DNB and 2,4-DNB ligands are non-emissive; however, nine of the complexes reported by Jassal et al. have shown a moderate amount of photoluminescence due to enhanced ligand-to-ligand charge transfer (LLCT) [82], facilitated by the rigid crystal structure of the complex. A list of selected 3,5- and 2,4-DNBA metal complexes is presented in Table 4. The M06-2X/6-31G** level of theory was employed to study monomeric ethylenediamine, 2,4-DNBA copper complexes. The molecular assembly and organization of these complexes led to the identification of a supramolecular 2D motif, where cooperative non-covalent interactions were analyzed through Hirshfeld surface analysis, in addition to their experimentally observed photoluminescence by considering a ligand-to-ligand charge transfer (LLCT) state, with additional minor contributions from metal-to-ligand charge transfer (MLCT) involving Cu(II) to 2,4-DNBA [95].

Theoretical investigations were conducted at the BP86-D3/def2-TZVP level of theory to examine the changes in the coordination sphere and noncovalent interactions in the solid state when pyridine (py) acts as an ancillary ligand. Two metallic complexes, [Cd_2_(µ-L)_2_(L)_2_(py)_4_] and [Zn(L)_2_(py)_2_], where L = 3,5-DNBA [82], were considered in the study. The results highlight the significance of lone pair-π and π-π stacking intermolecular stabilizing interactions in the solid state. Furthermore, these complexes exhibit fluorescence, with emission attributed to intraligand charge transfer processes at 377 nm and 374 nm for the Cd(II) and Zn(II) complexes, respectively.

Fonseca et al. synthesized the [Co(3,5-DNBA)_2_] complex and carried out several recrystallization experiments using various solvents [2]. The analysis of X-ray diffraction data revealed that the compounds obtained from H_2_O, CH_3_OH, EtOH:(CH_3_)_2_CO, and DMSO resulted in mononuclear molecules, whereas those recrystallized in acetone and MeCN produced trinuclear molecules. To explain these differences, DFT calculations were performed. The results indicated that the crystal growth in each solvent can be considered as a thermodynamic process favored by the crystallization energy (ΔE_recrys_), which is calculated as ΔE_recrys_ = E_fcrys_ − E_icrys_, where E_fcrys_ represents the energy of the final crystal structure, while E_icrys_ corresponds to the sum of the energies of the initial crystals and solvent molecules.

Another explanation based on ligand field theory suggests that the molecular orbitals of the solvents, which act as Lewis bases, can coordinate with the Co(II) center. To assess the strength of these interactions, the energy difference between the highest occupied molecular orbital (HOMO) of the ligands and the singly occupied molecular orbitals (SOMOs) of Co(II) (the three 3d orbitals) was evaluated using the B3LYP-DKH-D3BJ/def2-TZVP level of theory, agreeing with the experimental findings.

Ibragimov et al. synthesized three monometallic complexes consisting of 3,5-DNBA and monoethanolamine (MEA) [42]. These complexes, with the formula [M(DNBA)_2_(MEA)_2_] (M = Cu(II), Ni(II), Co(II)), can exist in different spin states. DFT calculations at the B3LYP/def2-TZVP and B3LYP/6-311G(d,p) levels of theory revealed that the most stable ground electronic states for the Cu, Ni, and Co complexes correspond to high-spin states, specifically a doublet, triplet, and quartet, respectively. Additionally, the mapped electrostatic potential surface (EPS) demonstrated that the electron-rich NO_2_ groups in the [M(DNBA)_2_(MEA)_2_] complexes can participate in non-covalent interactions with various molecules or contribute to crystal packing through intermolecular interactions. Furthermore, the analysis of the EPS revealed a strong electron deficiency near the NH_2_ in MEA, making it prone to hydrogen bonding. The most negative EPS values were located in the surroundings of the oxygen atoms of the NO_2_ substituents, indicating that they can act in intermolecular interactions as potential electron donors. The [Co(3,5-DNBA)_2_] was used to prepare six different derivatives, [Co(L)_2_(3,5-DNBA)_2_], L = bidentate and tridentate pyrazole ligands [74]. Despite the ability of DNB metallic complexes to form intermolecular interactions in the solid state, single-crystal X-Ray structures of these complexes could not be achieved. However, their proposed structures were obtained employing DFT calculations using the BP86-D3/TZ2P methodology, showing a substantial increase in the molecular dipole moment from the parent [Co(3,5-DNB)_2_] complex due to the inclusion of the pyrazole ligands. This is related to the higher antifungal activity that complexes present against *Candida albicans*, in comparison with the free ligands, which exhibit a dose-dependent antifungal activity. 

Moreover, DFT calculations enable the characterization of the electron transfer mechanism in cocrystals involving 3,5-DNBA units and [{Zn(L)(DMF)_4_}·2BF_4_]_α_ (L = N^2^,N^6^-di(pyridin-4-yl)naphthalene-2,6-dicarboxamide), in addition to the resulting HOMO-LUMO and bandgap values [98]. The results show that the LUMO position can be tuned to accept the incoming electron from π-electron-rich ligands. Furthermore, the authors explained the observed charge transfer fluorescence-quenching phenomenon based on DFT calculations, where an exception to the expected trend is obtained in one case, suggesting additional quenching mechanisms. Additionally, the role of the co-ligands 3,5-dimethylpyrazole (dmpz) and bis(3,5-dimethyl-1H-pyrazol-1-yl)methane in Zn(II) and Cu(II) DNBA monomeric and trimeric complexes was studied using the M06 DFT functional [5], demonstrating the structural versatility in the solid state of DNBA metallic complexes. This is due to the rationalization in the formation of a trinuclear species experimentally found only for Cu(II).

DFT calculations have been utilized to facilitate the strategic development of rare-earth coordination complexes based on DNBA units, enabling the assessment of the properties of the resulting molecular material [58]. Geometrical and electronic properties were evaluated for [Ln(3,5 DNBA)_3_(H_2_O)_2_]_n_ species, with Ln = La(III) and Gd(III), by Hazra et. al., denoting the paramagnetic character of the 4f1 and 4f7 ground states, respectively. The obtained HOMO-LUMO gap suggests further kinetic stability for such species and low reactivities. In the resulting species, the LUMO + 1 to LUMO + 4 manifold is near degeneracy, which is of 4f character, and thus centered at the lanthanide atom. In addition, the HOMO-3 and HOMO-2 are centered at the oxygen 2p orbital from the peripheral -NO_2_ substituents from the DNBA derivative, suggesting a ligand-to-metal charge transfer upon electronic excitation.

Also, the importance of Non-Covalent Interactions in the 2,4- and 3,5-Dinitrobenzoate Heterometallic {Eu2Cd2} complexes was afforded by analyzing the rotational energies around the carboxyl and nitro groups around the benzoic moiety at the PBE/def2tzvp level of theory [97]. Despite the fact that different binuclear lanthanide complexes have been reported for 3,5-dinitrobenzoates, [Ln_2_(Phen)_2_(3,5-Nbz)_6_] (Ln(III) = Dy, Tb, Gd, Eu, Sm, Ce) [58,99,100,101,102], and polymeric derivatives containing 2,4-dinitrobenzoates, such as [Ce_2_(phen)_2_(2,4-DNBA)_6_]_n_ and [Ln(2,4-DNBA)_3_(phen)]_n_ (Ln(III), x = Sm, Eu, Gd, Tb, Dy, Er), to our knowledge, none of them have been theoretically characterized.

In this context, several binuclear lanthanides 3,5-dinitrobenzoates with N,N-dimethylaniline (DMA) [103], 1,2-phenylenediamine [57], and N,N,N′,N′-tetramethyl-p-phenylenediamine were synthesized by Koroteev et al. [104], which showed interesting charge transfer properties that give rise to magnetism according to the amount of charge transfer, where Dysprosium complexes even show a single-molecule magnetic behavior. Moreover, coordination polymers based on bridged dicyanamido-metal(II) building units were synthesized, with their structural and magnetic characteristics rationalized via DFT calculations, by Mautner et al. [105]. In addition, the evaluation of gas adsorption performance for small molecules such as CO_2_, O_2_, N_2_, and CH_4_ provides valuable insights for the understanding of the adsorption affinity towards incoming molecules, bonded by noncovalent interactions, of coordination polymers organized as pillared layers involving Cu(II) metal centers. The information gained from the ωB97XD and B3LYP levels of theory is useful for the rational design of novel adsorption materials with higher selectivity and adsorption capacities, as reported by Meza-Morales et.al. [106]. 

## 6. Conclusions

In summary, DNB corresponds to a versatile ligand with a great variety of coordination modes from a general perspective. The presence of nitrobenzoate-derived ligands in the complexes contributes to their coordination versatility and can generate different mono–di-trinuclear complexes that can be tested toward microorganisms. This potential activity will contribute to the development of promising drugs that may have fewer side effects than currently used drugs. The reported crystal structures for these compounds with metallic centers such as Na, K, Cs, Ag, and Tl show coordination bonds involving carboxylate and nitro groups, as suggested by electrostatic potential analysis. However, it is interesting that in the reported crystal structures of compounds containing the DNB ligand with Ln and 3d transition metals, only the carboxylate group tends to participate in the coordination bonding (except for very few examples). This behavior seems predominant in the great majority of structures, suggesting that nitro groups prefer to participate in intermolecular interactions to help stabilize the packing. Despite this preference, in very sensible magnitudes, the synthesis and crystallization routes are factors of great influence over the structure of the products. Low changes in reaction conditions and the basicity hardness of crystallization solvents can drive the structural assembly in different paths, inducing different compounds and crystal structures with a high impact on the properties. Starting from theoretical calculations, electronic and bonding properties can be evaluated, which are useful for a quantitative structure–activity relationship (QSAR) and are required to further understand the overall chemical modifications of the complexes, which is useful for achieving improved design strategies. Particularly, bonding properties are useful to rationalize the electron-withdrawing capabilities led by the inclusion of the metal center upon the formation of coordination compounds using DNB. Overall, this review highlights the potential of coordination compounds, particularly those using nitrobenzoic acid and its derivatives. By exploring the recent literature and analyzing crystallographic data, the importance of understanding the electronic properties of these compounds through crystal structure elucidation and computational methods is highlighted. 

## 7. Outlook

This work offers a comprehensive perspective on the use of coordination complexes based on nitrobenzoic acid (NBA) and its derivatives in biological applications. It focuses on drug resistance in infectious diseases and cancer treatment. The need for developing new compounds with improved properties and fewer side effects is highlighted. Therefore, future perspectives may include the synthesis and evaluation of new complexes, along with a deeper understanding of their mechanisms of action and crystal structure. Coordination complexes based on NBA and its derivatives are presented as promising approaches to address these challenges, emphasizing their biological activity and structural behavior. The mechanisms through which these complexes exert their biological activity, such as the chelate effect and modulation of lipophilicity, enable them to penetrate cell membranes and affect essential functions of microorganisms and cancer cells.

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
