# Peer review of "Biological Activity of Complexes Involving Nitro-Containing Ligands and Crystallographic-Theoretical Description of 3,5-DNB Complexes"

_ijms, 2024, doi:10.3390/ijms25126536_

Round 1

Reviewer 1 Report

Comments and Suggestions for Authors

Thank you very much for choosing me as a potential reviewer for the Manuscript Title “Biological activity of complexes involving nitro-containing ligands and crystallographic-theoretical description of 3,5-DNB complexes. This work presents In this review, we summarize recent literature on coordination  compounds based on nitrobenzoic acid (NBA) as a ligand, its derivatives, and other nitro-containing ligands, which are widely employed owing to their versatility, displaying a range of coordination modes, such as monodentate, bidentate, bidentate bridging, and ionic, standing in different coordination sites through carboxylate and nitro groups.

The review needs minor revision. Detailed explanation should be given to the following comments:

Comments

1.      It is essential to further carefully proofread the review for grammar, punctuation, and spelling errors.

2.      The review is generally well-organized and clear. However, consider breaking down some long sentences for improved readability.

3.      Abstract part needs major correction. Aim and outcome of the work should be highlighted. No need of detailed discussion.

4.      The introduction needs some improvements. Can you provide more insight into the rationale behind selecting the specific ligand and metal ions for complexation in your study?

5.      Add statistical for data??

6.      The conclusions provided in the review are generally well-supported by the data and analyses presented. However, consider expanding on the implications of the research findings and how they contribute to the broader field of inorganic chemistry?

This research work shows promise, and with some revisions and clarifications, it will make a valuable contribution to the field. Addressing the points mentioned above will enhance the overall quality and readability of the paper.

Comments on the Quality of English Language

 It is essential to further carefully proofread the review for grammar, punctuation, and spelling errors.

Author Response

MDPI IJMS Editorial Office

May 24, 2024

First of all, I would like to thank the referees for the in-depth review the document. Again, after some time we have strived to improve our research work, based on the comments and suggestions of the referees. This letter shows the reviewers' comments made in the previous submission of the manuscript. The comments are shown explicitly and the way in which we responded each one of them to improve the manuscript.

Reviewer 1

Thank you very much for choosing me as a potential reviewer for the Manuscript Title “Biological activity of complexes involving nitro-containing ligands and crystallographic-theoretical description of 3,5-DNB complexes. This work presents In this review, we summarize recent literature on coordination compounds based on nitrobenzoic acid (NBA) as a ligand, its derivatives, and other nitro-containing ligands, which are widely employed owing to their versatility, displaying a range of coordination modes, such as monodentate, bidentate, bidentate bridging, and ionic, standing in different coordination sites through carboxylate and nitro groups.

The review needs minor revision. Detailed explanation should be given to the following comments:

Comments

  1. It is essential to further carefully proofread the review for grammar, punctuation, and spelling errors.

R/ The manuscript was sent for English revision.

  1. The review is generally well-organized and clear. However, consider breaking down some long sentences for improved readability.

R/ The wording of long sentences was improved.

  1. Abstract part needs major correction. Aim and outcome of the work should be highlighted. No need of detailed discussion.

R/ The abstract was written in a more general way.

  1. The introduction needs some improvements. Can you provide more insight into the rationale behind selecting the specific ligand and metal ions for complexation in your study?

R/ The introduction didn't delve into any specific selection process for ligands or metal ions. Instead, it provided a broad overview of the importance of coordination compounds based on nitrobenzoic acid (NBA) and other nitro-containing ligands. It primarily aimed to highlight the potential of coordination compounds in these areas and the importance of understanding their structural and electronic properties.

  1. Add statistical for data??

R/ No need to add statistics for data.

  1. The conclusions provided in the review are generally well-supported by the data and analyses presented. However, consider expanding on the implications of the research findings and how they contribute to the broader field of inorganic chemistry?

R/ The mistake was properly corrected.

This research work shows promise, and with some revisions and clarifications, it will make a valuable contribution to the field. Addressing the points mentioned above will enhance the overall quality and readability of the paper.

Comments on the Quality of English Language

It is essential to further carefully proofread the review for grammar, punctuation, and spelling errors.

Reviewer 2 Report

Comments and Suggestions for Authors

In this review, the authors reported the recent literature on coordination compounds based on nitrobenzoic acid (NBA) as a ligand, its derivatives, and other nitro-containing ligands, which are widely employed owing to their versatility, displaying a range of coordination modes, such as monodentate, bidentate, bidentate bridging, and ionic, standing in different coordination sites through carboxylate and nitro groups. Also, considering the importance of the elucidation of crystalline structures and computational calculations that allow to improve the characterization and the knowledge of electronic properties in the coordination complexes. Several nitro-containing compounds are reviewed on their antifungal and antibacterial capabilities. Where the metal center is able to modulate the bio-reduction abilities of the coordination compound. Considering the great structural diversity, an analysis of crystallographic data is presented unravelling the coordination preferences and the most effective crystallization methods to grow crystals of good quality. I recommend this review to be accepted after minor revision. Below are my specific comments which need to be considered before final acceptance.

1.     The abstract is a crucial part in any publications and it should be providing a concise summary of the study. So, I suggest the authors to rewrite it and avoiding the linguistic errors like in P1, line 29 and 30 “Several nitro-containing compounds are reviewed on their capabilities antifungal, antibacterial, where the metal center is able to modulate the bioreduction abilities of the coordination compound” which should be written as “Several nitro-containing compounds are reviewed on their antifungal and antibacterial capabilities. Where the metal center is able to modulate the bio-reduction abilities of the coordination compound”.

2.     In the introduction section, there is a deficiency of references such as P1, line 40, it must have some references for the different applications of nitro-containing ligands and their corresponding complexes. Also, P1, line 41, “Particularly, high number of reports make particular reference to the antimicrobial and anticancer activities of those complexes” NO references are mentioned, and so on. Therefore, kindly the authors must mention the required references.

3.     Schemes and figures captions must be clear and linguistically right.

4.     It will be better if the authors mentioned the spectroscopic techniques in P2, line 49, with the corresponding citation.

5.     Please bold font of subtitles, schemes, figures, tables, and compound numbers in the context.

6.     The abbreviations of 4-chloro-3-nitrobenzoic acid, P4, line 143, should be (4-Cl-3-NBA) to mention the position of nitro group as in 4-nitrobenzoic acid (4-NBA). The same issue in P8, figure 7, “2-chloro-5-nitro-benzoic acid”.

7.     P5, figure 2, the circles highlighted the nitro groups are not fitted it. Please make it fitted.

8.     In P6, the anticancer activity of NBA derivative complexes section, the proper cancer cell line assay and its inhibitory scheme, the binding with the DNA, and inhibit cancer activity didn’t mentioned in the context, please mention it. Moreover, there is no anticancer activity mentioned for specific cells of theses complexes, except breast cells.

9.     P8, line 259, “Complex 43” should be 24, it’s mistakenly written.

10.  P11, figure 9 and 10 quality need to enhanced.

11.  Why there is no self-assembly solution method mentioned in the context? and it’s one of the most commonly used methods as mentioned in P12, line 401.

12.  The superscript of the compound formula mentioned in table 2, P14, line 463 is wrong and not mentioned in the abbreviation list under the table.

13.  The room temperature abbreviations in table 3, P15, are different. The authors should use one abbreviation.

14.  P15, subtitle 5, not clear and should be modified.

Comments on the Quality of English Language

Generally, there are too much linguistically mistakes, which must be corrected.

Author Response

MDPI IJMS Editorial Office

May 24, 2024

First of all, I would like to thank the referees for the in-depth review the document. Again, after some time we have strived to improve our research work, based on the comments and suggestions of the referees. This letter shows the reviewers' comments made in the previous submission of the manuscript. The comments are shown explicitly and the way in which we responded each one of them to improve the manuscript.

Reviewer 2

In this review, the authors reported the recent literature on coordination compounds based on nitrobenzoic acid (NBA) as a ligand, its derivatives, and other nitro-containing ligands, which are widely employed owing to their versatility, displaying a range of coordination modes, such as monodentate, bidentate, bidentate bridging, and ionic, standing in different coordination sites through carboxylate and nitro groups. Also, considering the importance of the elucidation of crystalline structures and computational calculations that allow to improve the characterization and the knowledge of electronic properties in the coordination complexes. Several nitro-containing compounds are reviewed on their antifungal and antibacterial capabilities. Where the metal center is able to modulate the bio-reduction abilities of the coordination compound. Considering the great structural diversity, an analysis of crystallographic data is presented unravelling the coordination preferences and the most effective crystallization methods to grow crystals of good quality. I recommend this review to be accepted after minor revision. Below are my specific comments which need to be considered before final acceptance.

  1. The abstract is a crucial part in any publications, and it should be providing a concise summary of the study. So, I suggest the authors to rewrite it and avoiding the linguistic errors like in P1, line 29 and 30 “Several nitro-containing compounds are reviewed on their capabilities antifungal, antibacterial, where the metal center is able to modulate the bioreduction abilities of the coordination compound” which should be written as “Several nitro-containing compounds are reviewed on their antifungal and antibacterial capabilities. Where the metal center is able to modulate the bio-reduction abilities of the coordination compound”.

R/ This line was corrected as suggested.

  1. In the introduction section, there is a deficiency of references such as P1, line 40, it must have some references for the different applications of nitro-containing ligands and their corresponding complexes. Also, P1, line 41, “Particularly, high number of reports make particular reference to the antimicrobial and anticancer activities of those complexes” NO references are mentioned, and so on. Therefore, kindly the authors must mention the required references.

R/ The required references were mentioned.

  1. Schemes and figures captions must be clear and linguistically right.

R/ Schemes and figure captions are corrected to be clear and linguistically correct.

  1. It will be better if the authors mentioned the spectroscopic techniques in P2, line 49, with the corresponding citation.

R/ The required references were mentioned.

  1. Please bold font of subtitles, schemes, figures, tables, and compound numbers in the context.

R/ Subtitles, diagrams, figures, tables and numbers composed in context were put in bold.

  1. The abbreviations of 4-chloro-3-nitrobenzoic acid, P4, line 143, should be (4-Cl-3-NBA) to mention the position of nitro group as in 4-nitrobenzoic acid (4-NBA). The same issue in P8, figure 7, “2-chloro-5-nitro-benzoic acid”.

R/ The abbreviation for 4-chloro-3-nitrobenzoic acid was changed to 4-Cl-3-NBA to mention the position of the nitro group. Similarly, 2-chloro-5-nitro-benzoic acid was changed to 2-Cl-5-NBA.

  1. P5, figure 2, the circles highlighted the nitro groups are not fitted it. Please make it fitted.

R/ In Figure 2 the circles highlighting the nitro groups of complexes 12 and 13 are already adjusted.

  1. In P6, the anticancer activity of NBA derivative complexes section, the proper cancer cell line assay and its inhibitory scheme, the binding with the DNA, and inhibit cancer activity didn’t mentioned in the context, please mention it. Moreover, there is no anticancer activity mentioned for specific cells of theses complexes, except breast cells.

R/ The anticancer activity of the NBA-derived complex section is similar to that of cisplatin as expressed in Scheme 3. Within the reports we found, a specificity of the cells used is not clarified.

  1. P8, line 259, “Complex 43” should be 24, it’s mistakenly written.

R/ The mistake was properly corrected.

  1. P11, figure 9 and 10 quality need to enhanced.

R/ We improve the resolution of Figures 9 and 10.

  1. Why there is no self-assembly solution method mentioned in the context? and it’s one of the most commonly used methods as mentioned in P12, line 401.

R/ Indeed, the self-assembly method is one of the most important synthesis paths in order to form these compounds. Following the level of importance, we described some examples from line 470 to 499, including the information in Table 3.

  1. The superscript of the compound formula mentioned in table 2, P14, line 463 is wrong and not mentioned in the abbreviation list under the table.

R/ The meaning of the abbreviations bbmi = 1,1′-(1,4-butanediyl)bis(2-methylbenzimidazole) and bix: 1,4-bis(imidazol-1-ylmethyl)benzene were added.

  1. The room temperature abbreviations in table 3, P15, are different. The authors should use one abbreviation.

R/ The mistake was properly corrected.

  1. P15, subtitle 5, not clear and should be modified.

R/ The title "Calculations of the structures of NBA-derived complexes" was changed to "Theoretical study of 3,5- and 2,4-dinitrobenzoate metal complexes."

Comments on the Quality of English Language

Generally, there are too much linguistically mistakes, which must be corrected.

Reviewer 3 Report

Comments and Suggestions for Authors

Article entitled “Biological activity of complexes involving nitro-containing ligands and crystallographic-theoretical description of 3,5-DNB complexes” contains information on coordination compounds based on nitrobenzoic acid, its derivatives, and other nitro-containing ligands. The methods of theoretical chemistry are an important advantage of this work. The work is well laid out and the figures are aesthetically presented. I have some comments on the manuscript to help improve it.

1) The authors need to correct the nomenclature of the ligand : there is 4-nitrobenzoic acid, but there should be : 4-nitro benzoate. The point is that in the structures of the complexes there is an anion and not an acid molecule (e.g. Fig. 2) 

2) Authors need to correct editorial errors, e.g., line 172 "The Complexes" 

3) Lines 176-177 the phrase "whereas tetracycline and free ligands are negative" is very laconic and colloquial, so it needs to be rewritten and clarified.

4) Missing to complete the work is the section "Future Prospects/Outlook."

5) Item "4. Coordination behavior of the 3,5-DNB ligand and crystallization methods of the complexes: a perspective from the crystal structures in the CCDC database."  The paper should be rewritten to refer to the etmata of the work, i.e. biological activity. Because aktualnei it deviates from the rest of the work.

6) It is important to mention the differences in the biological activity of cisplatin and transplatin. 

7) Reference should also be made to the work: Pranczk, J., Jacewicz, D., Wyrzykowski, D., & Chmurzyński, L. (2014). Complex compounds of platinum (II) and palladium (II) as anticancer drugs. Methods for determining cytotoxicity. Current Pharmaceutical Analysis, 10(1), 2-9. 

Comments on the Quality of English Language

Minor editing of English language required.

Author Response

MDPI IJMS Editorial Office

May 24, 2024

First of all, I would like to thank the referees for the in-depth review the document. Again, after some time we have strived to improve our research work, based on the comments and suggestions of the referees. This letter shows the reviewers' comments made in the previous submission of the manuscript. The comments are shown explicitly and the way in which we responded each one of them to improve the manuscript.

Reviewer 3

Article entitled “Biological activity of complexes involving nitro-containing ligands and crystallographic-theoretical description of 3,5-DNB complexes” contains information on coordination compounds based on nitrobenzoic acid, its derivatives, and other nitro-containing ligands. The methods of theoretical chemistry are an important advantage of this work. The work is well laid out and the figures are aesthetically presented. I have some comments on the manuscript to help improve it.

1) The authors need to correct the nomenclature of the ligand : there is 4-nitrobenzoic acid, but there should be : 4-nitro benzoate. The point is that in the structures of the complexes there is an anion and not an acid molecule (e.g. Fig. 2) 

R/ The nomenclature of the ligand was changed from 4-nitrobenzoic acid to 4-nitro benzoate.

2) Authors need to correct editorial errors, e.g., line 172 "The Complexes" 

R/ The mistake was properly corrected.

3) Lines 176-177 the phrase "whereas tetracycline and free ligands are negative" is very laconic and colloquial, so it needs to be rewritten and clarified.

R/ The mistake was properly corrected.

4) Missing to complete the work is the section "Future Prospects/Outlook."

R/ An outlook section has been added.

5) Item "4. Coordination behavior of the 3,5-DNB ligand and crystallization methods of the complexes: a perspective from the crystal structures in the CCDC database."  The paper should be rewritten to refer to the etmata of the work, i.e. biological activity. Because aktualnei it deviates from the rest of the work.

R/ The subject of the review is about NBA-derived complexes: their possible application around biology, the coordination behavior of the 3,5-DNB ligand within this the methods of crystallization of the complexes and the calculations of the structures of NBA-derived complexes will be expanded.

6) It is important to mention the differences in the biological activity of cisplatin and transplatin. 

R/ We never refer to transplatin since it is not as effective as cisplatin in the treatment of cancer, its study is important to better understand the properties and mechanisms of action of platinum compounds, but not as a carcinogenic agent.

7) Reference should also be made to the work: Pranczk, J., Jacewicz, D., Wyrzykowski, D., & Chmurzyński, L. (2014). Complex compounds of platinum (II) and palladium (II) as anticancer drugs. Methods for determining cytotoxicity. Current Pharmaceutical Analysis, 10(1), 2-9. 

R/ We do not consider adding a reference in the review.

Comments on the Quality of English Language

Minor editing of English language required.
